# An Optimization-Based Framework for Adversarial Defence of Graph Neural Networks Via Adaptive Lipschitz Regularization

## Abstract

Graph Neural Networks (GNNs) have exhibited exceptional performance across diverse application domains by harnessing the inherent interconnectedness of data. However, the emergence of adversarial attacks targeting GNNs poses a substantial and pervasive threat, compromising their overall performance and learning capabilities. While recent efforts have focused on enhancing GNN robustness from both data and architectural perspectives, more attention should be given to overall network stability in the face of input perturbations. Prior methods addressing network stability have routinely employed gradient normalization as a fundamental technique. This study introduces a unifying approach, termed as AdaLip, for adversarial training of GNNs through an optimization framework that leverages the explicit Lipschitz constant. By seamlessly integrating graph denoising and network regularization, AdaLip offers a comprehensive and versatile solution, extending its applicability and enabling robust regularization for diverse neural network architectures. Further, we develop a provably convergent iterative algorithm, leveraging block majorization-minimization, graph learning, and alternate minimization techniques to solve the proposed optimization problem. Simulation results on real datasets demonstrate the efficacy of AdaLip over state-of-the-art defence methods across diverse classes of poisoning attacks. On select datasets, AdaLip demonstrates GCN performance improvements of up to 20% against modification attacks and approximately 10% against injection attacks. Remarkably, AdaLip achieves a similar performance gain on heterophily graph datasets.

## 1 Introduction

Graph Representation Learning, a emerging research field, focuses on extracting meaningful representations from graph-structured data. This has gained prominence due to Graph Neural Networks (GNNs), a class of neural networks capable of processing graph data and capturing both structural and semantic properties (Zhou et al., 2019; Xie et al., 2022; Yang et al., 2022; Kipf & Welling, 2016). However, GNNs, as a generalization of Deep Neural Networks, suffer from low interpretability, making them susceptible to adversarial attacks. Their performance heavily depends on the training graph, making them sensitive to structural changes and minor perturbations (Zhou et al., 2019; Xie et al., 2022; Yang et al., 2022; Kipf & Welling, 2016).

Adversarial attacks are those malicious perturbations of the graph structure or features that can fool the GNNs into making incorrect predictions (Dai et al., 2018; Jin et al., 2021; Geisler et al., 2021; Scholten et al., 2022). For instance, in community detection within graph data, adversarial attacks manifest as a strategic manipulation of the graph structure to conceal or distort the underlying community structure. These networks often exhibit suboptimal performance in scenarios where there exists a misalignment between the probability distributions of training and testing datasets, a condition frequently attributed to data corruption or inconsistent data collection methodologies during the test phase (Karniadakis et al., 2021; Zhou et al., 2022; Li et al., 2023).

In recent times, there has been a growing body of research focused on investigating the vulnerability of GNNs and devising diverse strategies to significantly enhance the robustness of GNNs against subtle perturbations (Entezari et al., 2020; Zhang & Zitnik, 2020; Xu et al., 2019; Zhang et al.,

2021; Loveland et al., 2021). Nevertheless, prevalent approaches can generally be classified into two principal categories: those grounded in a data-centric perspective of the problem and those emphasizing adversarial training. The data-centric viewpoint on network robustness is intrinsically reliant on the underlying assumptions of attack models. In contrast, the pursuit of robustness through adversarial training necessitates a complete overhaul of the underlying base network architecture. A significant challenge lies in the adaptability of these techniques across a wide array of network architectures. On the contrary, this research explores methods for enhancing the robustness of training across diverse architectural models by inherently minimizing the likelihood of failure, quantified through its stability coefficient.

In pursuit of stabilizing the neural network training on a noisy/perturbed topological structure for better generalization, we encounter the following challenges: first, adaptive structure learning for improved performance and further jointly ensuring the network does not become too sensitive towards input perturbation. To this end, we propose a unified optimization-based framework for handling the robustness and stability issues associated with GNN. Initially, we characterize adversarial connections as topological alterations that impact the smoothness of vertices, and we aim to refine the graph structure to eliminate such links (Wu et al., 2019). Secondly, inspired by the systems stability analysis using Lischitz properties, we propose a adaptive regularization to control the stability of a network. Enforcing Lipschitz continuity while training a machine has demonstrated its significant influence on better stability and generalization (Bartlett et al., 2017; Cranko et al., 2021; Cisse et al., 2017).

The efficacy of our proposed model in mitigating various forms of adversarial attacks is substantiated through comprehensive empirical evaluations conducted across a diverse set of real-world graph datasets. Evaluations are carried out on our proposed formulation under both graph modification and node injection types of poisoning attacks. Consistently, the performance of existing state-of-the-art defense techniques is surpassed, demonstrating the robustness and superiority of our model in safeguarding against adversarial threats.

The paper is organized as follows: Section 2 provides a comprehensive review of GNNs, adversarial attacks on GNNs, and defense strategies. Section 3 introduces the necessary notations and formally defines our proposed framework. In Section 4, we present the optimization algorithm for solving the proposed optimization problem. Section 5 reports experimental results, demonstrating the effectiveness of our framework, followed by concluding remarks in Section 6.

## 2 BACKGROUND AND RELATED WORK

In this section, a concise review of the working principles of GNN is provided, and the details of adversarial attacks are introduced. Additionally, in this discussion, light is shed on defense mechanisms developed for enhancing the adversarial robustness of GNN.

### 2.1 GRAPH NEURAL NETWORKS

The GNN primarily requires two types of operations, i) message passing and ii) feature aggregation, iteratively for several rounds. consider an undirected graph $\mathcal{G} = (\mathcal{V}, \mathcal{E})$, where $\mathcal{V}$ is the vertex set, and $\mathcal{E}$ is the edge set. The edge set is a subset of the ordered pair $\mathcal{V} \times \mathcal{V}$. Each vertex $n \in \mathcal{V}$ has the initial representation $r_n \in \mathbb{R}^d$, where $d$ is the dimension of the feature space. The representation of the embedding of vertex $n$ at the $(k+1)^{th}$ round of GNN operation is described as following

$$r_n^{(k+1)} = \mathcal{U}\left(r_n^{(k)}, \sum_{m \in \mathcal{N}(n)} \mathcal{M}\left(r_n^{(k)}, r_m^{(k)}\right)\right) \tag{1}$$

where $\mathcal{M}(\cdot)$ is a function that computes the message based on the current node representations of the neighborhood and $\mathcal{U}(\cdot)$ is a function that computes the embedding update based on the current node representation and the aggregated message. Based on different selection of the functions $\mathcal{U}(\cdot)$ and $\mathcal{M}(\cdot)$ different graph neural network architectures are derived in the literature.

### 2.2 ADVERSARIAL ATTACKS ON GRAPH NEURAL NETWORKS

Perturbation attacks on GNNs can be categorized into targeted, non-targeted, and random attacks based on the objectives and strategies employed by adversaries. In a targeted attack, the adversary

aims to manipulate the GNN's predictions by focusing on a specific set of nodes or classes Zügner et al. (2018). The attacker aims to craft perturbations that lead to incorrect predictions or misclassification for the targeted nodes or classes. A non-targeted attack, on the other hand, aims to disrupt the overall performance of the GNN without specific target nodes or classes in mind Sun et al. (2020). The attacker introduces perturbations that cause the GNN to make incorrect predictions across the entire graph, undermining the model's effectiveness and accuracy. In contrast, a random attack involves introducing random or arbitrary perturbations to the input data without a specific target or objective. The attacker introduces random noise or alterations to disrupt the GNN's predictions without a specific goal. Furthermore, there is another category of attacks that involves the injection of new malicious nodes into a network, without modifying the existing graph structure, while preserving the original edges and attributes (Zou et al., 2021; Sun et al., 2020).

In general, the attackers goal is to solve the following optimization problem:

$$\min_{\mathcal{G}_p \in \mathbb{P}(\mathcal{G}, \text{idx})} \mathcal{L}_a(f_{\theta^\star}(\mathcal{G}_p)) \text{ s.t } \theta^\star = \arg\min_\theta \mathcal{L}_{GNN}(f_\theta(\mathcal{G}_p)) \tag{2}$$

where, $\mathcal{G}$ and $\mathcal{G}_p$ are the original and perturbed graphs respectively. $\mathbb{P}$ is a perturbation function, $\mathcal{L}_a$ is the loss of attacker. The overall goal of the attacker is to hamper the generalization performance of the GNN.

## 2.3 ADVERSARIAL DEFENCE METHODS

As the issue of GNN robustness gains increasing prominence, there has been significant advancement in the pursuit of effective defense strategies (Zügner & Günnemann, 2019; Bojchevski & Günnemann, 2019). Impelled by the observation that targeted attacks elevate the rank of the adjacency matrix, a prior study introduced GCN-SVD, a method focused on mitigating the impact of such attacks by identifying and removing malicious edges associated with targeted nodes (Entezari et al., 2020). Following advancements in low-rank adjacency matrix learning, ProGNN was introduced, applying the same principle while concurrently learning the GNN model (Jin et al., 2020). Another defense approach leverages the spectral properties inherent in graph-structured data to facilitate the learning of a denoised graph (Wu et al., 2019; Dai et al., 2022; Chang et al., 2021). The recently proposed RWL-GNN framework adopts a similar concept, combining the principles of smooth structured graph learning and GNN parameter learning into a unified single objective (Runwal et al., 2022).

Despite these significant advancements in defense strategies, it is noteworthy that while these methods have focused on denoising the graph and adapting the GNN model to the denoised graph, none of them have addressed the critical aspect of ensuring the stability of the GNN model during training, particularly in the face of perturbations. This crucial issue of adapting the graph data to make the GNN model stable in handling perturbation challenges remains an open research frontier.

## 3 PROBLEM FORMULATION

The overall objective of learning a stable hypothesis $\theta$ as GNN parameters under the constraint of learning a denoised graph structure can be formulated as the general optimization problem:

$$\arg\min_{\theta \in \mathbb{S}_\theta} \mathcal{L}_{GNN}(\theta, \phi^*, \chi, y_l) \quad \text{s.t. } \phi^* = \arg\min_{\phi \in \mathcal{S}_\phi} \mathcal{L}_c(\phi, \chi, \phi_p) \tag{3}$$

where, $(\chi, y_l)$ corresponds to the training feature-label pairs and $\mathcal{L}_c(\cdot)$ is the loss corresponding to the noise removal. The set $\mathbb{S}_\theta$ contains all possible stable hypothesis functions mapping from the domain set to the labelled set. The constraint set on the graph structure is defined as, $\mathcal{S}_\phi = \left\{ \phi : \phi = \phi^T, \phi_{ij} \le 0, \phi_{ii} = -\sum_{j \ne i} \phi_{ij} \right\}$. Fundamentally, the optimization problem can be understood as learning a stable GNN hypothesis function over a cleaned surrogate $\phi^\star$ of the given noisy graph $\phi_p$. The problem can be rewritten as a joint objective

$$\min_{\theta \in \mathbb{S}_\theta, \phi \in \mathcal{S}_\phi} \mathcal{L}_{GNN}(\theta, \phi, \chi, y_l) + \lambda \mathcal{L}_c(\phi, \chi, \phi_p) \tag{4}$$

The objective in (4) can accommodate various architectures based on the message passing scheme (1), as well as different structure relearning schemes, whether they involve rank-constrained methods or

rely on spectral properties. However, a critical challenge in optimizing the objective is parameterizing the set $\mathbb{S}_\theta$ to guide the learning process, given that estimation may be NP-hard.

## 3.1 STABILITY OF GRAPH NEURAL NETWORKS

**Lipschitz Functions.** A mapping $f : \mathbb{R}^n \mapsto \mathbb{R}^m$ is globally $\mathbb{L}$-Lipschitz over any domain space $\mathcal{D}$ if there exists a constant $\mathbb{L} \geq 0$ such that

$$\|f(x_1) - f(x_2)\| \leq \mathbb{L} \|x_1 - x_2\|, \quad \forall x_1, x_2 \in \mathcal{D} \tag{5}$$

where, the smallest feasile value of $\mathbb{L}$ satisfying (5) is termed the Lipschitz constant of the mapping function $f(\cdot)$. In the context of stability analysis, when examining both the original point and a perturbed point represented as $x_2 = x_1 + \delta$ in (5), the Lipschitz constant provides an upper bound for the change in the functional value concerning the perturbation $\delta$.

Stability in the GNN relative to perturbations in the normalized graph adjacency matrix can be formalized as in (Gama et al., 2020; Kenlay et al., 2021)

$$\sup_{x \in \mathcal{X}} \|\theta(x, \Delta) - \theta(x, \Delta_p)\| \in \mathcal{O} \left( \|\Delta - \Delta_p\| \right) \tag{6}$$

where, $\Delta$ is the normalized adjacency matrix of the original graph and $\Delta_p$ is the normalized adjacency matrix for the perturbed graph. Normalized adjacency matrix for a graph with degree matrix $D$ and adjacency matrix $A$ is expressed as $\Delta = D^{-1/2}(A + I)D^{-1/2}$.

**Theorem 1.** *Consider the normalized adjacency matrices $\Delta$ and $\Delta_p$, where $\Delta_p$ represents the perturbed version of $\Delta$. We define their normalized difference as $\|\Delta - \Delta_p\| \overset{\text{def}}{=} \|E\|$. Let $f_\theta$ be a GCN hypothesis function with $L$ layers, with embeddings denoted as $f_\theta(x, \Delta)$ for the original graph and $f_\theta(x, \Delta_p)$ for the perturbed graph. The upper bound on the difference between these embeddings is as follows:*

$$\|\theta(x, \Delta) - \theta(x, \Delta_p)\|_F \leq \sqrt{d}L \|E\|_2 \prod_{l=1}^{L} \left\| \theta^{(l)} \right\| \tag{7}$$

*where, $\theta^{(l)}$ corresponds to the GCN parameters of the $l^{th}$-layer.*

*Proof.* A detailed proof is provided in the Appendix A. □

Conventional networks lack constraints on their parameters, enabling them to exhibit exceptional discriminative capabilities thanks to the inherent piecewise non-linearity in their architecture. However, constraining the search space, as outlined in (6), can render the network less responsive to high-frequency components within the data. Balancing a small Lipschitz constant with the goal of achieving high discriminative ability is often considered a challenging trade-off. Moreover, estimating the exact Lipschitz bound corresponding to a given network also makes enforcing such restrictions difficult.

One effective method for enforcing Lipschitz constraints on a network during training involves normalizing the parameters of each layer using Projected Gradient Descent (PGD). This is defined as $\theta_{(k+1)}^{(l)} \leftarrow \Pi_{\mathcal{P}} \left( \theta_{(k)}^{(l)} - \eta \nabla_{\theta^{(l)}} \mathcal{L}_{GNN} (\theta, \phi, \chi, y_l) \right)$, where $\Pi$ represents a projection function that projects the parameters into the space $\mathcal{P}$. Nevertheless, we contend that relying solely on the suggested PGD approach in solving (4) could potentially undermine the overall discriminative capabilities of GNNs and result in suboptimal defense strategies due to its lack of adaptivity to the input data. Therefore, we reformulate the optimization problem (4) introducing an explicit adaptive regularization mechanism to accommodate a realizable Lipschitz bound during the training.

## 3.2 PROPOSED ADAPTIVE LIPSCHITZ REGULARIZATION FRAMEWORK

In the context of stability, the bound presented in (7) should be as small as possible. However, to learn a network with a high discriminative ability and with a realizable bound, we propose an adaptive Lipschitz regularization (AdaLip)

$$\min_{\theta, \phi \in \mathcal{S}_\phi} \mathcal{L}_{GNN}(\theta, \phi, \chi, y_l) + \lambda \mathcal{L}_c(\phi, \chi, \phi_p) + \gamma \|\Delta - \Delta_p\| \prod_{l=1}^{L} \left\| \theta^{(l)} \right\| \tag{8}$$

**Lemma 1.** *Let $\Delta_p$ represent the perturbed adjacency matrix, and $\Delta_{(k)}$ denote the optimal adjacency matrix corresponding to the optimal graph Laplacian $\phi_{(k)}$ at the $k^{th}$ iteration while solving (8). For a suitable value of $\beta$, it holds that $\|\Delta - \Delta_p\| \leq \beta \|\Delta - \Delta_{(k)}\|$. Proof is in the Appendix B. The parameter $\beta$ is absorbed in the regularization parameter $\gamma$.*

Formally, we seek to iteratively find $\theta_{(k+1)}$ and $\phi_{(k+1)}$ that can minimize the following objective function

$$(\theta_{(k+1)}, \phi_{(k+1)}) = \underset{\theta, \phi \in \mathcal{S}_\phi}{\arg\min} \ \mathcal{L}_{GNN}(\theta, \phi, \chi, y_l) + \lambda \mathcal{L}_c(\phi, \chi, \phi_p) + \gamma \|\Delta - \Delta_{(k)}\| \prod_{l=1}^{L} \left\|\theta^{(l)}\right\| \quad (9)$$

## 3.3 FURTHER DISCUSSION

In this subsection, we contextualize the AdaLip formulation within the framework of strategies pertaining to the fortification of GNNs against adversarial threats and the enhancement of their stability. Our proposed optimization-based framework serves as a unifying structure that both encompasses and extends several previously proposed methods, effectively positioning them as special cases within our theoretical framework.
**ProGNN:** Set $\gamma = 0$ and $\mathcal{L}_c$ as low-rank approximation objective.
**RWL-GNN:** Set $\gamma = 0$ and $\mathcal{L}_c$ as learning smooth graph Laplacian.
**GCN-SVD:** Solve in two-stage by setting $\gamma = 0$ and $\mathcal{L}_c$ as low-rank approximation objective.
**LipRelu:** Take $\lambda = 0$ (Jia et al., 2023).

Figure 1: An illustration for the motivation of AdaLip.

**ERNN:** Take $\lambda = 0$, and enforce $\gamma \|\Delta - \Delta_{(k)}\| \prod_{l=1}^{L} \|\theta^{(l)}\| = 1$ (Zhao et al., 2021).
At a high level, one might draw parallels between our AdaLip framework and (Gama et al., 2020). However, a clear distinction lies in usability: AdaLip is a plug-and-play method compatible with any GNN architecture, while the latter necessitates enforcing specific properties in graph filters. Moreover, our AdaLip framework is used to enhance GNN robustness against deliberately crafted attacks, distinguishing it from the latter, which primarily addresses inference errors.

We empirically compare various GNN training strategies on perturbed graph structures, evaluating Lipschitz bounds and discriminative capabilities using the CiteSeer dataset (see Figure 1). In the case of AdaLip, we adopt the Graph Convolutional Network (GCN) architecture with a smooth graph learning objective. Vanilla GCN without defense mechanisms and regularization performs poorly. Introducing $\ell_2$ regularization improves the Lipschitz bound but maintains subpar discriminative ability. RWL-GNN achieves strong discriminative performance but at the cost of a significantly higher Lipschitz bound. In contrast, AdaLip strikes a balance, maintaining a smaller Lipschitz bound while achieving top discriminative performance. We also explore the adaptivity of our AdaLip framework by omitting the graph denoising component (AdaLip w/o graph denoising i.e., $\lambda = 0$), which is similar to LipRelu (Jia et al., 2023). This configuration results in a smaller Lipschitz bound but is suitable only for low perturbations, akin to (Gama et al., 2020).

**Remark 1.** While we focused on node classification using GCN as base architecture in this work, our proposed method is versatile and adaptable to various GNN architectures and learning settings.

**Remark 2.** The results in Figure 1 underscore the significance of data adaptivity within AdaLip. Both low-rank adjacency matrix approximation and learning graph Laplacian with reduced eigen energy prove effective for this purpose. While implementing either approach can introduce substantial

computational complexity, it is important to note that this complexity is not a bottleneck specific to AdaLip. Existing literature provides methods to mitigate computational challenges associated with both approaches. However, a comprehensive analysis of these methods is beyond the scope of this paper.

## 4 OPTIMIZATION ALGORITHM FOR ADAPTIVE LIPSCHITZ REGULARIZATION

To tackle the challenging multi-block non-convex optimization problem (9), we employ two optimization strategies. First, a two-stage approach separates graph denoising from GNN parameter optimization, ensuring computational efficiency without sacrificing results. Additionally, we utilize an efficient iterative algorithm based on Block Successive Upper Bound Minimization (BSUM) to simultaneously optimize both variables. This algorithm iteratively updates individual blocks, striking a balance between variable interdependencies for convergence.

In this work, we have developed an iterative algorithm applicable to GNNs with a general architecture, where the denoising objective is defined as the minimization of the eigenenergy. Without loss of generality, the adaptive Lipschitz regularization in (9) can be equivalently replaced by a logarithmic counterpart. The optimization problem will be

$$(\theta_{(k+1)}, \phi_{(k+1)}) = \underset{\theta, \phi \in \mathcal{S}_\phi}{\arg\min} \ \mathcal{L}_{GNN}(\theta, \phi, \chi, y_l) + \alpha \left\| \phi - \phi_p \right\|_F^2 + \beta \text{Tr}(\chi^T \phi \chi)$$

$$+ \gamma \log \left( \sqrt{d} \left\| \Delta - \Delta_{(k)} \right\|_F^2 \right) + \gamma \sum_{l=1}^{L} \log \left\| \theta^{(l)} \right\|_2^2 \qquad (10)$$

To simplify the minimization of the aforementioned problem, we adopt linear operators, as in (Kumar et al., 2020), to transform the complex variable matrix $\phi$ into a vector.

**Lemma 2.** *By defining linear operators $\mathcal{L}$, $\mathcal{A}$ and respective adjoint operators $\mathcal{L}^\star \mathcal{A}^*$, the objective function of (10) can be transformed to*

$$(\theta_{(k+1)}, \omega_{(k+1)}) = \underset{\theta, \omega \geq 0}{\arg\min} \ \mathcal{L}_{GNN}(\theta, \omega, \chi, y_l) + \alpha \left\| \mathcal{L}\omega \right\|_F^2 - \mathcal{L}^\star (2\alpha\phi_p^T - \beta\chi^T\chi)^T \omega$$

$$+ \gamma \log \left( \sqrt{d} \left\| \mathcal{A}\omega - \mathcal{A}\omega_{(k)} \right\|_F^2 \right) + \gamma \sum_{l=1}^{L} \log \left\| \theta^{(l)} \right\|_2^2 \qquad (11)$$

*Proof.* Please refer to Appendix C for detailed analysis. □

### 4.1 TWO STAGE APPROACH: ADALIP.2

**Graph Denoising Stage** The objective for pre-processing the noisy graph will be

$$\omega_{(k+1)} = \underset{\omega \geq 0}{\arg\min} \ p(\omega) = \alpha \left\| \mathcal{L}\omega \right\|_F^2 - \mathcal{L}^\star (2\alpha\phi_p^T - \beta\chi^T\chi)^T \omega + \gamma \log \left( \sqrt{d} \left\| \mathcal{A}\omega - \mathcal{A}\omega_{(k)} \right\|_F^2 \right)$$

(12)

To solve the non-convex optimization problem we employ the procedure of Majorization-Minimization. The objective function $p(\omega)$ is $L_\omega$-smooth, hence, first order Taylor series expansion can be used as the surrogate function. See Appendix D. The majorized surrogate at $\omega_{(k)}$ will be

$$p(\omega|\omega_{(k)}) = p(\omega_{(k)}) + \left\langle \nabla_\omega p(\omega_{(k)}), \omega - \omega_{(k)} \right\rangle + \frac{L_\omega}{2} \left\| \omega - \omega_{(k)} \right\|^2 \qquad (13)$$

Next, in the minimization step the following problem will be solved

$$\omega_{(k+1)} = \underset{\omega \geq 0}{\arg\min} \ p(\omega|\omega_{(k)}) \qquad (14)$$

Update rule for the problem (14) will be

$$\omega_{(k+1)} = \left( \omega_{(k)} - \frac{1}{L_\omega} \nabla_\omega p(\omega|\omega_{(k)}) \right)_+ \qquad \text{where,} \ (a)_+ = \max\{0, a\} \qquad (15)$$

**Learning GNN parameters** The optimization problem corresponding to learning of GNN parameters will be

$$\min_{\theta} \ \mathcal{L}_{GNN}(\theta, \omega^{\star}, \chi, y_l) + \gamma \sum_{l=1}^{L} \log \left\| \theta^{(l)} \right\|_2^2 \tag{16}$$

where, $\omega^{\star}$ is optimal solution from the graph denoising part. The problem is solved using gradient descent, update rule of which is

$$\theta_{(k+1)} = \theta_{(k)} - \eta \nabla_{\theta} \left( \mathcal{L}_{GNN}(\theta_{(k)}, \omega^{\star}, \chi, y_l) + \gamma \sum_{l=1}^{L} \log \|\theta_{(k)}^{(l)}\|_2^2 \right) \tag{17}$$

where $\eta$ is the learning rate and in our experiments the gradient is obtained using AutoGrad function in the PyTorch.

## 4.2 JOINT OPTIMIZATION APPROACH: ADALIP.J

In this section, we introduce an iterative algorithm that utilizes the block successive upper bound minimization (BSUM) technique (Razaviyayn et al., 2013; Sun et al., 2016). Collecting both the variables $\mathbb{V} = \{\omega \geq 0, \theta\}$, we propose a block-based majorization-minimization (MM) algorithm that updates one variable at a time while holding the other variable constant.

**Update of GNN parameters** To update the GNN parameter $\theta$, we fixed the variable $\omega$ and solve the following optimization problem

$$\min_{\theta} \ \mathcal{L}_{GNN}(\theta, \omega, \chi, y_l) + \gamma \sum_{l=1}^{L} \log \left\| \theta^{(l)} \right\|_2^2 \tag{18}$$

The approximate solution to above problem is obtained using one step gradient descent similar to the update rule (17). However, one can run the updates for several iterations.

**Update of graph weights** Treating the variable $\omega$ as variable while fixing the GNN parameters, we obtain the following sub-problem

$$\omega_{(k+1)} = \arg\min_{\omega \geq 0} \ q(\omega) = \mathcal{L}_{GNN}(\theta, \omega, \chi, y_l) + \alpha \left\| \mathcal{L}\omega \right\|_F^2 - \mathcal{L}^{\star}(2\alpha\phi_p^T - \beta\chi^T\chi)^T\omega$$
$$+ \gamma \log \left( \sqrt{d} \left\| \mathcal{A}\omega - \mathcal{A}\omega_{(k)} \right\|_F^2 \right) \tag{19}$$

By assuming the GNN loss Lipschitz smooth, we obtain the following majorant surrogate of $q(\omega)$ at $\omega_{(k)}$ will be $q(\omega|\omega_{(k)}) = q(\omega_{(k)}) + \langle \nabla_{\omega} p(\omega_{(k)}), \omega - \omega_{(k)} \rangle + \frac{L_q}{2} \left\| \omega - \omega_{(k)} \right\|^2$

See supplementary material for detailed analysis. Next, in the minimization step the following problem will be solved

$$\omega_{(k+1)} = \arg\min_{\omega \geq 0} \ q(\omega|\omega_{(k)}) \tag{20}$$

Update rule for the problem (20) will be

$$\omega_{(k+1)} = \left( \omega_{(k)} - \frac{1}{L_q} \left[ \nabla_{\omega} \mathcal{L}_{GNN}(\theta, \omega_{(k)}, X, y_l) + 2\alpha\mathcal{L}^{\star} \left( \mathcal{L}\omega_{(k)} + \alpha\phi_p^T - \beta X^T X \right) + \gamma \frac{2\mathcal{A}^{\star}(\mathcal{A}\omega_{(k)}) - \omega_{(k)}}{\left\| \mathcal{A}\omega - \mathcal{A}\omega_{(k)} \right\|_F^2} \right] \right)_+ \tag{21}$$

We conduct an ablation study concerning the hyperparameter $\gamma$ in the Appendix. The algorithm is summarized in Appendix E.

**Theorem 2.** *The sequence* $\{\omega_{(k)}, \theta_{(k)}\}$ *generated by the algorithm converges to the set of Karush-Kuhn-Tucker (KKT) points of problem (19).*

*Proof.* Detailed proof is provided in the Appendix E. □

## 5 EXPERIMENTS

In this section, we validate the effectiveness of our proposed algorithm through a comprehensive set of experiments conducted on the Cora, CiteSeer (CS), Polblogs, Wisconsin, and Chameleon datasets. Additional results on the PubMed dataset are presented in the Appendix. We benchmark the proposed method against the following baselines: GCN(Kipf & Welling, 2016), GCN-SVD(Entezari et al., 2020), ProGNN(Jin et al., 2020), RGCN(Zhu et al., 2019), RWL-GNN(Runwal et al., 2022), and LipRelu (Jia et al., 2023). While we have tuned the hyperparameters of all methods following the guidelines provided by the respective authors, it is important to note that we conducted training for a fixed 400 epochs. Details of all experiments is provided in the Appendix G.

### 5.1 MODIFICATION ATTACK

**Setup.** In this subsection, we assess two graph modification attacks, Nettack and Metattack, within a poisoning setting. These evaluations follow the same setup as outlined in the ProGNN paper (Jin et al., 2020). Metattack operates as a global attack model, while Nettack is designed as a targeted attack model. In the case of Metattack, we introduce perturbations ranging from 0% to 25%, with a step size of 5%. Meanwhile, for Nettack, we perform 5 perturbations for all target nodes in the test data.

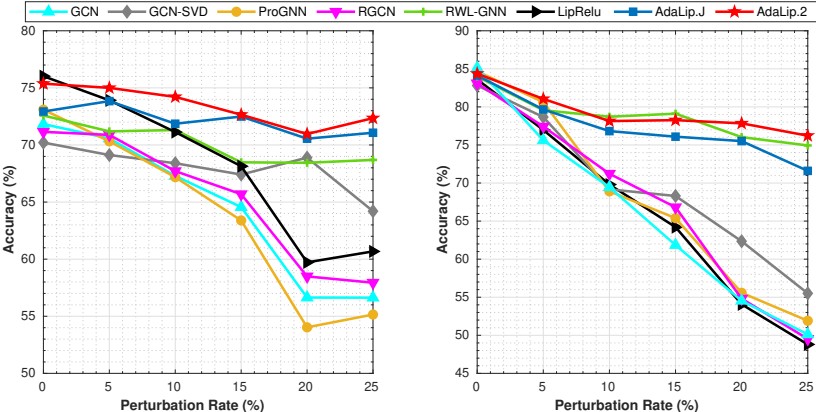

Figure 2: Node classification accuracy (%) under different perturbation rates of Metattack. Left: CiteSeer Dataset. Right: Cora Dataset.

Table 1: Node classification accuracy (%) under Nettack with 5 perturbation per target. Accuracy on clean graphs corresponding to each method is mentioned in the brackets.

| Dataset | GCN | GCN-SVD | ProGNN | RWL-GNN | AdaLip |
|---|---|---|---|---|---|
| CS-*Nettack* | 50.79 (71.83) | 69.83 (70.21) | 66.03 (73.09) | 73.84 (72.58) | **76.19 (75.37)** |
| Cora-*Nettack* | 57.11 (83.61) | 63.00 (82.81) | 61.30 (84.67) | 72.84 (83.98) | **74.04 (84.36)** |
| Polblogs-*Nettack* | 93.04 (96.56) | 94.58 (97.81) | 93.78 (97.21) | 94.16 (96.72) | **95.04 (97.04)** |

**Results.** In Figure 2, we present node classification accuracy results on the Cora and CS datasets for each defense model, specifically against Metattack. Our experiments utilize both optimization approaches outlined in Section 4. The results demonstrate that the AdaLip framework consistently outperforms the baseline methods across most scenarios. However, it is worth noting that in this experiment, the two-stage approach surpasses the joint method, possibly due to the approximate solution of (18) using a one-step gradient descent. Hence, we report the accuracy of the two-stage approach in subsequent experiments. Under Nettack, AdaLip outperforms all baseline methods in classification performance (see Table 1). Lipschitz bound analysis is presented in Figure 1. ProGNN needs to be trained for 1000 epochs to get competitive results (See Appendix).

## 5.2 INJECTION ATTACK

**Setup.** We also evaluate AdaLip in the context of injection attacks, where the original graph remains unperturbed, but malicious nodes with attributes are introduced near the target nodes. Specifically, we employ TDGIA (Zou et al., 2021), known for its potency in the KDD CUP dataset, for these injection attacks. The attacks are initially conducted on a surrogate GCN model, and subsequently, the poisoned graph is transferred to the defense models. For both the Cora and CS datasets, we maintain a maximum allowable limit of 500 nodes and 1000 edges.

Table 2: Node classification accuracy (%) under TDGIA injection attack. Accuracy on clean graphs corresponding to each method is mentioned in the brackets.

| Dataset | GCN | GCN-SVD | ProGNN | RWL-GNN | AdaLip |
|---------|-----|---------|--------|---------|--------|
| CS-*TDGIA* | 66.20 (71.83) | 49.39 (70.21) | 62.78 (73.09) | 68.23 (72.58) | **73.64 (75.37)** |
| Cora-*TDGIA* | 69.46 (83.61) | 50.39 (82.81) | 69.87 (84.67) | 75.53 (83.98) | **81.74 (84.36)** |

**Results.** Table 2 presents the performance of defense methods against TDGIA-based graph poisoning attacks. Sensitivity to adversarial nodes and edges can significantly impact overall performance. The results demonstrate that AdaLip notably enhances the performance of GCN on both Cora and CS datasets compared to the baseline methods. Techniques such as ProGNN, RWL-GNN, and AdaLip, which incorporate noise removal objectives, exhibit robustness against these attacks. The improved resilience of the AdaLip framework against injection attacks can be attributed to its additional adaptive Lipschitz regularization, which constrains the embedding differences between clean and poisoned graphs. In contrast, GCN-SVD performs less effectively than the vanilla GCN, as noise removal is conducted independently of the model's learning process.

## 5.3 PERFORMANCE ON HETEROPHILY GRAPHS

We assess the performance of the AdaLip framework on two heterophily datasets: Chameleon and Wisconsin. Given that the current AdaLip variant employs a feature smoothness objective similar to both ProGNN and RWL-GNN, which contradicts the heterophily assumption in these experiments, we have set the regularization constant $\beta \to 0$ in (10). For ProGNN and RWL-GNN, the respective regularizations were also removed from their formulations.

Table 3: Node classification accuracy (%) under Nettack (5) and Metattack (10%).

| Dataset | GCN | ProGNN | RWL-GNN | AdaLip |
|---------|-----|--------|---------|--------|
| Chameleon-*Nettack* | 38.16 | 53.82 | 61.56 | **63.09** |
| Chameleon-*Metattack* | 33.19 | 35.62 | 35.18 | **36.12** |
| Wisconsin-*Nettack* | 57.11 | 61.30 | 68.84 | **70.04** |
| Wisconsin-*Metattack* | 49.59 | 56.26 | 68.56 | **72.73** |

Table 3 reports the node classification accuracy of defence methods on both datasets. AdaLip achieves the highest discriminative performance among the three defence schemes, demonstrating its applicability to heterophily datasets. The strong performance of AdaLip with the GCN backbone suggests its potential for defending heterophilic-specific GNN architectures.

## 6 CONCLUSION

The discriminative ability of GNNs often diminishes in the presence of noisy or perturbed data. We introduce an optimization-based framework that enables the utilization of an explicit Lipschitz bound as adaptive regularization while simultaneously minimizing the combined loss of GNN training and graph denoising. Employing the two derived algorithms for optimization, our framework consistently achieves competitive performance across diverse attack scenarios on various real-world datasets. Future research will focus on a comprehensive analysis of different objectives and GNN architectures within the AdaLip framework.

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

## A PROOF OF THEOREM 1

For the GCN the embeddings at the $l^{th}$ layer is given as

$$X^{(l)} = \sigma\left(\Delta X^{(l-1)}\theta^{(l)}\right) \tag{1}$$

where $\theta^{(l)}$ are the GCN weights at the $l^{th}$ layer. For a $L$-layer GCN depicted as $f_\theta$ the notion of stability can be extended as

$$\|f_\theta(x,\Delta) - f_\theta(x,\Delta_p)\|_F = \left\|X^{(L)} - X_p^{(L)}\right\|_F = \left\|\sigma(\Delta X^{(L-1)}\theta^{(L)}) - \sigma(\Delta_p X_p^{(L-1)}\theta^{(L)})\right\|_F \tag{2}$$

Since, most activation functions have unit Lipschitz constant

$$\left\|\sigma(\Delta X^{(L-1)}\theta^{(L)}) - \sigma(\Delta_p X_p^{(L-1)}\theta^{(L)})\right\|_F \leq \left\|\Delta X^{(L-1)}\theta^{(L)} - \Delta_p X_p^{(L-1)}\theta^L\right\|_F \tag{3}$$

For two general matrices $P$ and $Q$, we have

$$\|PQ\|_F \leq \|P\|_F \|Q\|_2 \tag{4}$$

Further,

$$\left\|\Delta X^{(L-1)}\theta^{(L)} - \Delta_p X_p^{(L-1)}\theta^L\right\|_F \leq \left\|\Delta X^{(L-1)} - \Delta_p X_p^{(L-1)}\right\|_F \left\|\theta^{(L)}\right\|_2 \tag{5}$$

Further using triangle inequality we have,

$$\left\|\Delta X^{(L-1)} - \Delta_p X_p^{(L-1)}\right\|_F \left\|\theta^{(L)}\right\|_2 \leq \left(\left\|\Delta X^{(L-1)} - \Delta_p X^{(L-1)}\right\|_F + \left\|\Delta_p X^{(L-1)} - \Delta_p X_p^{(L-1)}\right\|_F\right)\left\|\theta^{(L)}\right\|_2 \tag{6}$$

Again using (4) and that $\Delta_p$ is normalized we get

$$\leq \left(\|E\|_2 \left\|X^{(L-1)}\right\|_F + \left\|X^{(L-1)} - X_p^{(L-1)}\right\|_F\right)\left\|\theta^{(L)}\right\|_2 \tag{7}$$

where, $E = \Delta - \Delta_p$. Since, $\left\|X^{(L)}\right\|_F \leq \left\|X^{(0)}\right\|\prod_{l=1}^{L}\left\|\theta^{(l)}\right\|_2$ and taking input data being $d$ dimensional normalized vectors i.e., $\left\|X^{(0)}\right\|_F = \sqrt{d}$

$$\leq \sqrt{d}\|E\|_2 \prod_{l=1}^{L}\left\|\theta^{(l)}\right\| + \sqrt{d}(L-1)\|E\|_2 \prod_{l=1}^{L}\left\|\theta^{(l)}\right\|$$

$$\left\|X^{(L)} - X_p^{(L)}\right\|_F \leq \sqrt{d}L\|E\|_2 \prod_{l=1}^{L}\left\|\theta^{(l)}\right\| \tag{8}$$

## B PROOF OF LEMMA 1

$$\|\Delta - \Delta_p\| = \left\|\Delta - \Delta_{(k)} + \Delta_{(k)} - \Delta_p\right\| \leq \left\|\Delta - \Delta_{(k)}\right\| + \left\|\Delta_{(k)} - \Delta_p\right\| \tag{9}$$

Now, since the norms are non-negative and we can use the following property for non-negative a,b, and c

$$\text{If } a \leq b + c \quad \text{then, } a \leq \zeta b \tag{10}$$

where, $\zeta \in \left[0, \frac{a}{b} + \frac{c}{b}\right]$

## C PROOF OF LEMMA 2

For the training of GNN we consider a given triplet $\mathcal{G} = (\mathcal{V}, \mathcal{E}, \omega)$. If the number of nodes, $|\mathcal{V}| = n$, then $\omega \in \mathbb{R}^n$ is the weight (adjacency) matrix with $\omega_{ij} = \omega_{ji} \geq 0$ and $\omega_{ii} = 0$. Degree matrix $D$ is a diagonal matrix with entries $D_{ii} = \sum_j \omega_{ij}$.

The Laplacian matrix, denoted as $\phi$, is an $n \times n$ symmetric matrix defined as $\phi = D - \omega$. The Laplacian matrix provides valuable insights into the graph's connectivity, as well as its spectral

properties. The graph Laplacian matrix is a positive semi-definite and $M$-matrix with non-positive off diagonal elements(Slawski & Hein, 2015). The vector $\mathbf{1} \in \mathbb{R}^n$, with each element as unity is the eigen vector corresponding to the smallest eigen value of the Laplacian matrix i.e., $\phi \mathbf{1} = 0$. Using the definition and properties of the graph Laplacian matrix we define a set

$$\mathcal{S}_\phi = \left\{ \phi \; : \; \phi = \phi^T, \phi_{ij} \leq 0, \phi_{ii} = -\sum_{j \neq i} \phi_{ij} \right\} \tag{11}$$

For simplification in the optimization problem we define linear operators same as in (Kumar et al., 2020).

**Definition 1.** The linear operator $\mathcal{L} : \mathbb{R}^{n(n-1)/2} \to \mathbb{R}^{n \times n}$ is defined as

$$[\mathcal{L}\omega]_{ij} = \begin{cases} \omega_{i+d_j} & i > j \\ [\mathcal{L}\omega]_{ji} & i < j \\ -\sum_{j \neq i} [\mathcal{L}\omega]_{ij} & i = j \end{cases} \tag{12}$$

where, $d_j = -j + \frac{i-1}{2}(2n-j)$

Using the above defnition of linear operator we can simplify (11) as

$$\mathcal{S}_\phi = \{\mathcal{L}\omega : \; \omega \geq 0\} \tag{13}$$

**Definition 2.** The adjoint of operator $\mathcal{L}^\star : \mathbb{R}^{n \times n} \to \mathbb{R}^{n(n-1)/2}$ is defined as

$$[\mathcal{L}^\star Y]_k = Y_{ii} - Y_{ij} - Y_{ji} + Y_{jj}, \; k = i - j + \frac{j-1}{2}(2n-j), \; i > j \text{ and } i, j \in \mathbb{Z}^+ \tag{14}$$

**Definition 3.** Linear operator $\mathcal{A} : \mathbb{R}^{n(n-1)/2} \to \mathbb{R}^{n \times n}$ is defined as

$$[\mathcal{A}\omega]_{ij} = \begin{cases} \omega_{i+d_j} & i > j \\ [\mathcal{A}\omega]_{ji} & i < j \\ 0 & i = j \end{cases} \tag{15}$$

where, $d_j = -j + \frac{i-1}{2}(2n-j)$

**Definition 4.** The adjoint operator $\mathbf{A}^\star : \mathbb{R}^{n \times n} \to \mathbb{R}^{n(n-1)/2}$ is defined as

$$[\mathcal{A}^\star Y]_k = Y_{ij} + Y_{ji}, \; k = i - j + \frac{j-1}{2}(2n-j), \; i > j \text{ and } i, j \in \mathbb{Z}^+ \tag{16}$$

Please refer (Kumar et al., 2020) for a more detailed analysis of the above defined linear operators.

By using the definition of Laplacian and Adjacency operators the optimization problem will be

$$(\theta_{(k+1)}, \omega_{(k+1)}) = \underset{\theta, \omega \geq 0}{\arg\min} \; \mathcal{L}_{GNN}(\theta, \omega, \chi, y_l) + \alpha \|\mathcal{L}\omega - \phi_p\|_F^2 + \beta \mathbf{Tr}(\chi^T \mathcal{L}\omega\chi)$$
$$+ \gamma \log\left(\sqrt{d}\|\mathcal{A}\omega - \mathcal{A}\omega_{(k)}\|_F^2\right) + \gamma \sum_{l=1}^L \log\left\|\theta^{(l)}\right\|_2^2 \tag{17}$$

By using the definition of the Laplace operator the problem (17) can be further simplified as

$$(\theta_{(k+1)}, \omega_{(k+1)}) = \underset{\theta, \omega \geq 0}{\arg\min} \; \mathcal{L}_{GNN}(\theta, \omega, \chi, y_l) + \alpha \|\mathcal{L}\omega\|_F^2 - \mathcal{L}^\star(2\alpha\phi_p^T - \beta\chi^T\chi)^T\omega$$
$$+ \gamma \log\left(\sqrt{d}\|\mathcal{A}\omega - \mathcal{A}\omega_{(k)}\|_F^2\right) + \gamma \sum_{l=1}^L \log\left\|\theta^{(l)}\right\|_2^2 \tag{18}$$

---

**Algorithm 1:** Joint Optimization Algorithm

---

1  **Input**   :Input graph $\mathcal{G} = (\mathcal{V}, \mathcal{E}, \mathcal{W})$, label matrix $Y$, number of layers $L$, regularization parameters $\alpha, \ \beta, \ \gamma$

     **Output** :$\theta, \ \omega$

2  **Initialize** $\omega_{(0)} \rightarrow \omega_n, \ \theta_{(0)} \rightarrow$ Randomly

3  **while** *stopping condition not met* **do**

4     |  **Graph weight update**

5     |  $\omega_{(k+1)} =$

        $\left( \omega_{(k)} - \frac{1}{L_q} \left[ \nabla_\omega \mathcal{L}_{GNN}(\theta, \omega_t, X, y_l) + 2\alpha \mathcal{L}^\star(\mathcal{L}\omega_t) + \mathcal{L}^\star(\alpha \phi_p^T - \beta X^T X) + \gamma \frac{2\mathcal{A}^\star(\mathcal{A}\omega_{(k)}) - \omega_{(k)}}{\left\| \mathcal{A}\omega - \mathcal{A}\omega_{(k)} \right\|_F^2} \right] \right)_+$

6     |  **GNN Parameter Update**

7     |  $\theta_{(k+1)} = \theta_{(k)} - \eta \nabla_\theta \left( \mathcal{L}_{GNN}(\theta_{(k)}, \omega, \chi, y_l) + \gamma \sum_{l=1}^L \log ||\theta_{(k)}^{(l)}||_2^2 \right)$

---

## D   Smoothness of the Objective Function

The optimization problem in consideration is

$$(\theta_{(k+1)}, \omega_{(k+1)}) = \underset{\theta, \omega \geq 0}{\arg \min} \ \mathcal{L}_{GNN}(\theta, \omega, \chi, y_l) + \alpha \left\| \mathcal{L}\omega \right\|_F^2 - \mathcal{L}^\star(2\alpha \phi_n^T - \beta \chi^T \chi)^T \omega$$

$$+ \gamma \log \left( \sqrt{d} \left\| \mathcal{A}\omega - \mathcal{A}\omega_{(k)} \right\|_F^2 \right) + \gamma \sum_{l=1}^L \log \left\| \theta^{(l)} \right\|_2^2 \qquad (19)$$

To render the smoothness analysis of the objective function tractable we assume the following

1. The GNN loss function $\mathcal{L}_{GNN}(\theta, \omega, \chi, y_l)$ is $L_1$-Lipschitz continuous i.e., the gradient is always upper bounded by a constant.
2. The layer wise norm of the GNN parameters will never converge to zero.

By analysis both the norm and trace terms in the objective are Lipschitz. By the above assumption $\sum_{l=1}^L \log \left\| \theta^{(l)} \right\|_2^2$ is continuously differentiable and hence Lipschitz smooth. Finally, until the convergence is attained the term $\log \left( \sqrt{d} \left\| \mathcal{A}\omega - \mathcal{A}\omega_{(k)} \right\|_F^2 \right)$ is also continuously differentiable. Hence, by the property of addition the overall function is Lipschitz smooth until the convergence.

## E   Algorithm and Convergence

The proposed optimization problem is

$$\min_{\theta, \omega \geq 0} \ \mathcal{L}_{GNN}(\theta, \omega, X, y_l) + \alpha \left\| \mathcal{L}w \right\|_F^2 - \mathcal{L}^\star(2\alpha \phi_n^T - \beta X^T X)^T \omega + \gamma \log \left( \sqrt{d} \left( \left\| \mathcal{A}\omega \right\|_F^2 - \omega^t \omega \right) \right) + \gamma \sum_{l=1}^L \log \left\| \theta^{(l)} \right\|_2^2$$
$$(20)$$

The Lagrangian function corresponding to the optimization problem (20)

$$L(\theta, \omega, \lambda \in \mathbb{R}) = \mathcal{L}_{GNN}(\theta, \omega, X, y_l) + \alpha \left\| \mathcal{L}\omega \right\|_F^2 - \mathcal{L}^\star(2\alpha \phi_n^T - \beta X^T X)^T \omega$$

$$+ \gamma \log \left( \sqrt{d} \left( \left\| \mathcal{A}\omega \right\|_F^2 - \omega^t \omega \right) \right) + \gamma \sum_{l=1}^L \log \left\| \theta^{(l)} \right\|_2^2 - \langle \lambda, \omega \rangle \qquad (21)$$

**KKT conditions for GNN parameters** $\theta$    From the stationarity condition

$$\nabla_\theta \mathcal{L}_{GNN}(\theta, \omega, X, y_l) + 2\gamma \sum_{l=1}^L \frac{\theta^{(l)}}{\left\| \theta^{(l)} \right\|_2^2} = 0 \qquad (22)$$

**KKT conditions for graph weights** $\omega$

1. From the primal feasibility condition: $\omega \geq 0$

2. From the dual feasibility condition: $\lambda \geq 0$

3. From the complementary slackness condition: $\langle \lambda, \omega \rangle = 0$

4. From the stationarity condition:

$$\nabla_\omega \mathcal{L}_{GNN}(\theta, \omega, X, y_l) + 2\alpha \mathcal{L}^\star(\mathcal{L}\omega) + \mathcal{L}^\star(\alpha\phi_n^T - \beta X^T X) + \gamma \frac{2\mathcal{A}^\star(\mathcal{A}\omega) - \omega^t}{\|\mathcal{A}\omega\|_F^2 - \omega^t\omega} = 0 \quad (23)$$

The update rule of $\theta$ is

$$\theta_{t+1} = \theta_t - \eta_\theta \left[ \nabla_\theta \mathcal{L}_{GNN}(\theta_t, \omega, X, y_l) + 2\gamma \sum_{l=1}^{L} \frac{\theta_t^{(l)}}{\|\theta_t^l\|_2^2} \right] \quad (24)$$

The update rule of $\omega$ is

$$\omega_{t+1} = \left( \omega_t - \eta_\omega \left[ \nabla_\omega \mathcal{L}_{GNN}(\theta, \omega_t, X, y_l) + 2\alpha \mathcal{L}^\star(\mathcal{L}\omega_t) + \mathcal{L}^\star(\alpha\phi_n^T - \beta X^T X) + \gamma \frac{2\mathcal{A}^\star(\mathcal{A}\omega_t) - \omega_t\omega}{\|\mathcal{A}\omega\|_F^2 - \omega_t} \right] \right)_+ \quad (25)$$

Since the weights of the graph $\omega$ are updated following the KKT conditions, by some scaled Lagrangain variable $\lambda$ we can conclude that $\omega_{(\infty)}$ satisfies the KKT conditions.
Hence, the sequence of solutions generated by the joint stage optimization converges to the KKT points of the problem (20).

## F    DETAILS OF EXPERIMENTAL SETUP

The experiments are performed on a workstation with, intel xeon CPU and 24 GB Nvidia GPU. We adopted a consistent experimental configuration similar to that of (Jin et al., 2020). Our chosen architecture is the two-layer GCN model. In each graph, we randomly allocated 10% for training, 10% for validation, and 80% for testing nodes. Hyperparameter tuning was conducted using the validation dataset.
We utilized the deeprobust library (Li et al., 2021) for both the implementation of attacks and the GCN architecture. The attack strategy aligns with the Pro-GNN approach outlined by (Jin et al., 2020). Specifically, for Nettack, target nodes were identified as those in the test set with a degree greater than 10. We varied the number of perturbations on each target node from 1 to 5 in increments of 1. As for Metattack, we explored perturbation levels ranging from 0% to 25% in 5% increments.

## G    ADDITIONAL RESULTS

In Table 4 we present the results of RGCN on Metattack. In Table 5, we present a comparison with ProGNN trained for 1000 epochs on CS dataset with Metattack.

Table 4: Results of RGCN.

| Ptb | CS-RGCN | Cora-RGCN |
|-----|---------|-----------|
| 0 | 71.15 $\pm$0.84 | 83.09$\pm$0.44 |
| 5 | 70.88$\pm$0.62 | 77.42$\pm$0.39 |
| 10 | 67.71$\pm$0.30 | 71.22$\pm$0.38 |
| 15 | 65.69$\pm$0.37 | 66.82$\pm$0.39 |
| 20 | 58.49$\pm$1.22 | 54.81$\pm$0.37 |
| 25 | 57.94$\pm$2.09 | 49.53$\pm$1.96 |

Table 5: Comparison with ProGNN trained for 1000 epochs.

| Ptb | ProGNN(1000 epochs) | AdaLip (400 epochs) |
|-----|---------------------|---------------------|
| 0   | 73.28               | 75.37               |
| 5   | 72.93               | 75                  |
| 10  | 72.51               | 74.21               |
| 15  | 71.03               | 72.65               |
| 20  | 69.02               | 70.96               |
| 25  | 68.95               | 72.35               |

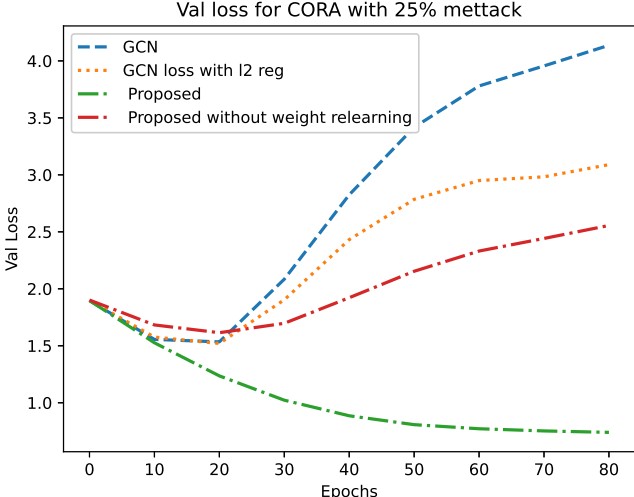

Figure 1: Validation loss vs. training iterations on Cora dataset.

### G.1 COMPARISON WITH OTHER PROPOSED APPROACHES FOR GENERALIZATION OF NEURAL NETWORK

Previously, several works have been pointing towards better generalization using decaying parameters (Krogh & Hertz, 1991; Liu et al., 2020). In this section we see that our approach cannot be viewed as straightforward application of previous works. To this end, on Cora and CS datasets after $25\%$ perturbation we train vanilla GCN, GCN with $\ell_2$-norm regularization, GCN with regularization of logarithmic norm of layer-wise network parameters, and our proposed two-stage algorithm. The results of this experiment are summarized in Figure 1 and Figure 2. Following both the results it is evident that by our proposed AdaLip framework provides better generalization compared to other methods. It is also evident that AdaLip with graph weight re-learning provides better generalization in comparison to just logarithmic norm regularization. Further, for more analysis we have used $20\%$ Meta attack perturbation on PubMed dataset and compared the performance of vanilla GCN, GCN with $\ell_2$-norm regularization and the proposed use of logarithmic norm regularization. In this method we have not done the pre-processing step due to lack of computational resources. The results presented in Figure 3 highlights that the regularization gives better generalization compared to vanilla GCN. Further, we perform an ablation for selection of the hyperparameter $\gamma$ (see Figure 4).

### G.2 ANALYSIS OF SMOOTHING EFFECT

As a Graph Neural Network (GNN) goes through multiple layers, each layer engages in message passing and adjusts the representations of nodes using information obtained from their neighboring nodes. This progressive propagation of information across the layers accentuates the effects of smoothing and diffusion. Consequently, nodes that were initially distant from each other in the graph tend to possess more alike representations after multiple layers of message passing, resulting in an intensified smoothing effect.

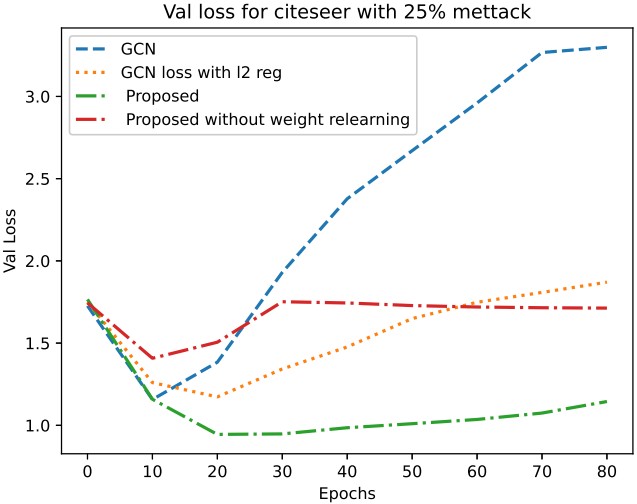

Figure 2: Validation loss vs. training iterations on CS dataset.

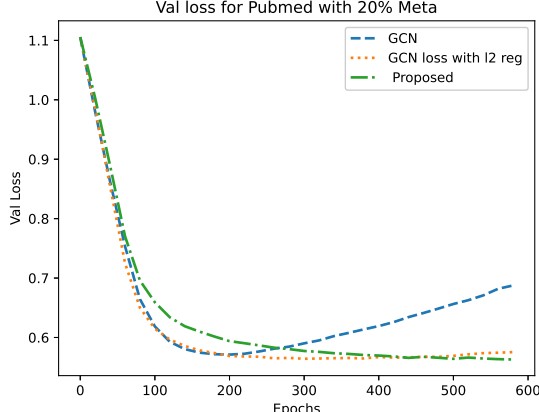

Figure 3: Validation loss vs. training iterations on PubMed dataset.

In this experiment we evaluate the performance of our proposed AdaLip framework to handle the oversmoothing effect. We compared Vanilla GCN and AdaLip on Cora and CS dataset without any perturbation. The results presented in Figure 5 and Figure 6 clearly highlights that the proposed AdaLip framework tackles the over-smoothing issue of GNN. Hence, the results validate that the proposed work leads to new paradigm of GNN training.

### G.3 LIPSCHITZ ANALYSIS OF ADALIP

Lipschitz constants of neural networks have been investigated in diverse deep learning contexts, including the assurance of adversarial robustness (Sokolić et al., 2017; Tsuzuku et al., 2018; Anil et al., 2019; Miyato et al., 2018). We provide the proof regarding the Lipschitz property of GNN. Let $(\chi, \Delta)$ be the input data and adjacency matrix and $Y$ the corresponding embeddings obtained from GNN with parameters $\theta$. Let the network is now fooled by adding perturbation $\delta$ in the data, the

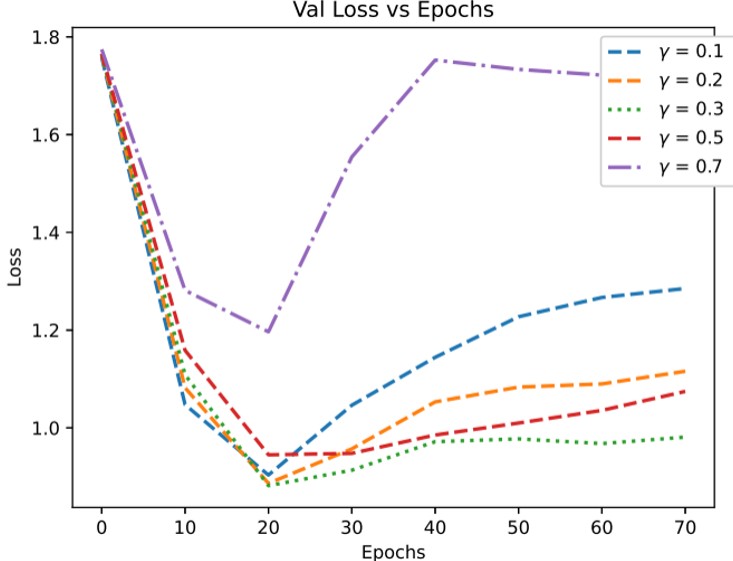

Figure 4: Validation loss vs. Epochs.

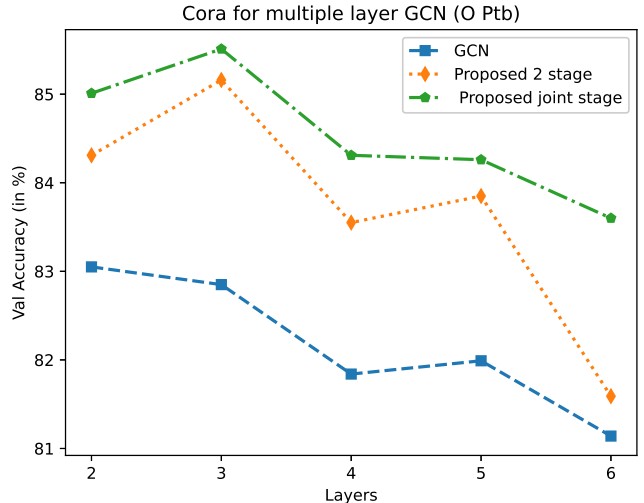

Figure 5: Validation accuracy vs. GNN layers on Cora dataset.

corresponding embedding will be $\tilde{Y}$.

$$Y = \sigma \left( \Delta \chi \theta \right) \tag{26}$$

$$\tilde{Y} = \sigma \left( \Delta \left( \chi + \delta \right) \theta \right) \tag{27}$$

$$\left\| Y - \tilde{Y} \right\| = \left\| \sigma \left( \Delta \chi \theta \right) - \sigma \left( \Delta \left( \chi + \delta \right) \theta \right) \right\| \tag{28}$$

$$\left\| Y - \tilde{Y} \right\| \leq \left\| \Delta \right\| \left\| \theta \right\| \left\| \chi - \left( \chi + \delta \right) \right\| \tag{29}$$

Figure 7 clearly highlights that the proposed AdaLip framework is adversarially robust.

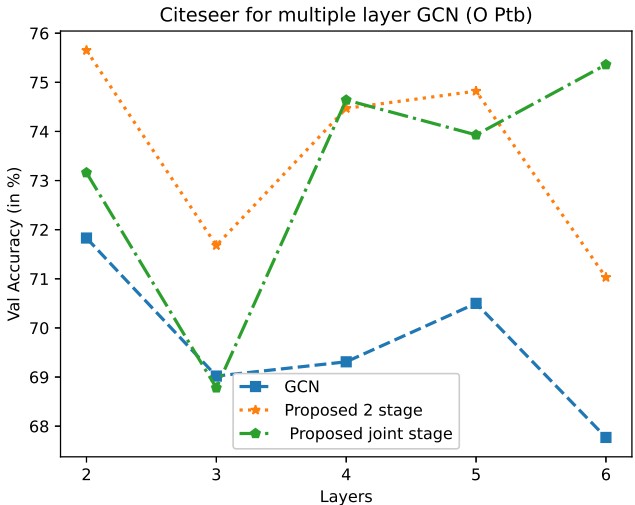

Figure 6: Validation accuracy vs. GNN layers on Citeseer dataset.

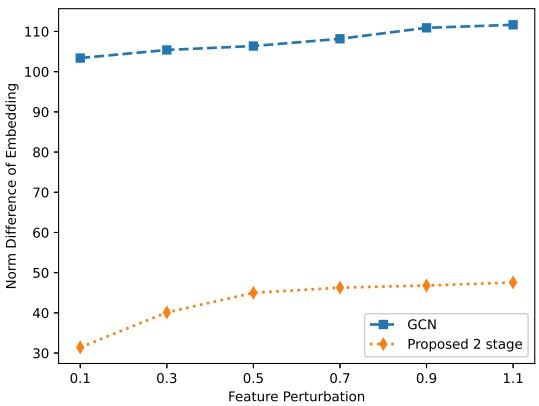

Figure 7: Analysis of the Lipschitz property for AdaLip.

