# OpenReview forum: "An Optimization-Based Framework for Adversarial Defence of Graph Neural Networks Via Adaptive Lipschitz Regularization"
_ICLR.cc/2024/Conference — Submitted to ICLR 2024_

### Official Review · Reviewer_C1i6 · 2023-10-25

**Soundness:** 3 good
**Presentation:** 2 fair
**Contribution:** 3 good
**Rating:** 5
**Confidence:** 4

**Summary:**

The presented paper proposed a unified framework based on optimization unfolding. The proposed framewrok combines methods based on updating the graph (denoising) and methods based on training (network regularization).

**Strengths:**

- The presented paper combines two worlds (graph denoising and regularization), and is flexible enough to allow different choice of components by using different loss function.
- The framework is derived from an optimization perspective, which provides good interpretability of the proposed method.
- Experiment results shows the effectiveness of the proposed method.

-----
I have read the author responce and decided to keep my rating unchanged.

**Weaknesses:**

- I found this paper rather sloppy in mathematics. In terms of:
    - Undefined synbols. See Questions.
    - Some synbols are overrided without explanation. For example, in eq.(6) $\theta$ is used to represent a function (the GNN model in my understanding), while in eq.(10) $\theta$ is used to represent the parameters of the model.
    - There are some imprecise terms.
        - From (3) to (4) the authors say "The problem (eq.(3)) can be rewritten as a joint objective (eq.(4))". I don't see how eq.(3) can be rewritten as eq.(4). Indeed eq.(4) is a relaxed version eq.(3), but they are not equavalent. Doing such relaxation usually requires some related properties of the two problems, e.g. they share the same global optima. For the eq.(3) to eq.(4), I don't see such a relation, at least the authors didn't mention any.
        - In section 4, there's a sentence "Without loss of generality, the adaptive Lipschitz regularization in (9) can be equivalently replaced by a logarithmic counterpart". I don't see how relacing a part of a equation by its logarithmic counterpart while leaving other parts unchanged is without loss of generality. It's likely eq.(9) and eq.(10) have different global optima. I would suggest the authors just add the log in the original definition eq.(9).
- For the two stage approach, I don't see why it can converge. The given theorem only proves the convergence of the joint optimization approach.
- The experiments are performed only on small graphs. I wonder what is the computational complexity of the proposed algorithm and if it is limited on small graphs?
- The proof of Theorem 1 is looks problematic. Overall, I think it's unlikely that the Lipschitz constant depends only on the parameters but not on the activation function and the structure of the GNN.
    - For the first inequal symbol in eq.(2) in the appendix, how is $\sigma$ disapeared? Doesn't this require $\sigma$ to be $1$-Lipschitz?
    - The proof of Theorem 1 assums a very specific structure of GNN (basically GCN), which mismatches the definition of GNN given in section 2.1. The authors should state it explicitly that it works only for a specific implementation of GNN in the statement of  Theorem 1. Alternatively, you can also assume $\mathcal M$ and $\mathcal U$ are Lipschitz continous and combine the Lipschitz constant of them into the bound.

**Questions:**

There are some undefined or unclear notations. Although I can guess most of them but it's better to define them clearly.

 - In eq.(1), what is $\mathcal N$?
 - In eq.(3), what is $\phi$? Also, based on the definition of $\mathcal S_{\phi}$, is $\phi \in \mathcal S_\phi$ just a graph Laplacian?
- In Theorem 1, is $E$ a scalar or vector? In the statement it says $|\Delta - \Delta_p| \overset{\text{def}}= |E|$, which makes it looks like a scalar, but in the equation it uses $\\|E\\|_2$, which makes it looks like a vector.
- In Lemma 2, what is $\omega$? Is it a scalar or vector?
- In eq.(21), what is $L_q$?

It's possible that I missed some definitions or assumptions. Feel free to point out if I missed something.

---

> ### Author Response · Authors · 2023-11-22
> **Response to Reviewer C1i6**
>
> Thanks for your detailed review. We are going to address your concerns one by one as follows:
>
> **Question: Clarification on the relaxation of problem (3) to (4)**
>
> Response: The authors agree that optimization problem (4) is relaxed version of (3). By solving (4) graph denoising and GNN training is done jointly, making the method adaptive.
>
> **Suggestion on using log based definition in (9)**
>
> Response: Thank you for the suggestion. We will revise the original definition according to the suggestion.
>
> **Question: Computational cost analysis**
>
> Response: The table shows runtime comparison of different methods for 400 epochs on Cora dataset.
>
> | **Method** | **Time (s)** |
> |------------|--------------|
> | GCN        | 5.87         |
> | ProGNN     | 770.56       |
> | AdaLip.J   | 201.89       |
> | AdaLip.2   | 56.29        |
>
> **Question: Convergence of two-stage approach**
>
> Response: We acknowledge that the two-stage algorithm aims to provide an approximate solution to the problem (11) with reduced computational resources. We intend to strike a balance between computational efficiency and solution quality. The results in the Table above demonstrate that our approach effectively achieves this objective, offering a compelling alternative that merits consideration. Indeed, the optimal solution obtained through the two-stage process may not strictly minimize the problem (11). However, in our analysis, we have observed that the solution serves as a robust proxy. The results align well with recent findings indicating the significance of the average degree of the input graph in the convergence of GNN [1]. Solving the problem (12) filters the adversarial connections, leading to a lower average degree in the data.
>
> **Question: Clarification on the proof of Theorem 1**
>
> Response: Thank you for the suggestion. We have modified the proof in the paper with all assumptions and details.
>
> **Question: Clarification of $\mathcal{N}$ in (1)**
> Response: It is the neighbourhood function.
>
> **Question: Clarification on $\phi$ in (3)**
> Response: It is the graph Laplacian
>
> **Question: Clarification on $\|\|E\|\|$ in Theorem 1**
> Response: Thank you for highlighting this. We have rectified in the paper.
>
> **Question: Clarification on $\omega$ in Lemma 2**
> Response: It is a vector. Detailed description is in Appendix C.
>
> **Question: Clarification on $L_q$ in (21)**
> Response: It is the Lipschitz constant of the function $q(\omega)$ defined in (19).
>
> [1]: Awasthi, P., Das, A., & Gollapudi, S. (2021). A convergence analysis of gradient descent on graph neural networks. Advances in Neural Information Processing Systems, 34, 20385-20397.

---

> ### Comment · Reviewer_C1i6 · 2023-11-22
>
> Thank the authors for their responce. My concerns has been partially addressed. However, there are still unaddressed issues. For the computational cost, what I expected was a computational complexity analysis, which can give us a sense of how fast the algorithm is in a direct way. The authors provided some experiment result on Cora. Although I appreciate the authors' efforts in conduction extra experiments, it is worth noting that Cora is a fairly small dataset and can not reflect the true complexity of an algorithm. Not to mention the proposed method looks quite slow on Cora. Based on this, I have decided to keep my current rating to this paper unchanged. I suggest the authors to rewrite and reorganize this paper with a more rigorous and consistent math notation and language and resubmit it.

---

### Official Review · Reviewer_uQo7 · 2023-10-29

**Soundness:** 2 fair
**Presentation:** 2 fair
**Contribution:** 2 fair
**Rating:** 3
**Confidence:** 4

**Summary:**

This work introduces an approach called AdaLip to improve GNN robustness. Specifically, authors first introduce an objective function based on the adaptive Lipschitz regularization, which aims to purify the graph topology and train robust GNNs. Subsequently, authors develop an iterative algorithm that is provably convergent for optimizing the objective function. Experimental results indicate that AdaLip outperforms a few defense baselines under the transfer attack setting.

**Strengths:**

- Authors have considered both graph modification and injection attacks.
- AdaLip has been evaluated on both homophily and heterophily datasets.

**Weaknesses:**

- Missing adaptive attack results. As shown by [1], most prior defense GNN methods can be easily broken by adaptive attacks, which are aware of the given defense method during attacking. Thus, it is very important to adaptively attack the proposed defense model to demonstrate its true robustness.
- Missing relevant defense models for evaluation. There are some prior methods (e.g., [2]) for defending on both homophily and heterophily datasets, which are not compared in this work.
- Improper claims. It is unclear why prior adversarial training methods (e.g., PGD) cannot be applied to different GNN architectures. Furthermore, since the authors exclusively focus on GCN as the GNN backbone in their experiments, their claim on the adaptability of AdaLip to various GNNs is less convincing. Additionally, the authors assert that PGD is not a suitable choice for solving Equation (4), but they provide no empirical results to support this claim.
- There is a lack of sensitivity analyses on $\alpha$ and $\beta$.
- The tightness of the upper bound in Theorem 1 is unclear.
- The paper writing can be further improved. Authors have introduced several terms without adequate explanations or definitions, some of which I've listed in the following questions.

[1]: Mujkanovic et al., “Are Defenses for Graph Neural Networks Robust?”, NeurIPS'22. \
[2]: Deng et al., “GARNET: Reduced-Rank Topology Learning for Robust and Scalable Graph Neural Networks”, LoG'22.

**Questions:**

- What's the "smoothness of vertices"? Do authors mean node feature smoothness?
- What does the "data adaptivity" mean?
- What's the definition of high-frequency components within the data? Do authors mean the Laplacian eigenvectors corresponding to the largest eigenvalues? If so, it's unclear to me why AdaLip can work on heterophily datasets since it is less responsive to those high-frequency components.
- What does the "adaLip w/o GD" mean in Figure 1?
- Remark 2 is somewhat unclear. Do the authors mean that AdaLip also employs low-rank approximation on the adjacency matrix? If so, could you please point out the equation in the paper that demonstrates this? Additionally, given that ProGNN also learns a low-rank adjacency matrix, why does Figure 1 illustrate the efficacy of the low-rank approximation?

---

> ### Author Response · Authors · 2023-11-22
> **Response to Reviewer uQo7**
>
> Thanks for your detailed review. We are going to address your concerns one by one as follows:
>
> **Question: Clarification on smoothness of vertices?**
>
> Response: By smoothness of vertices, the authors mean smoothness of node-level attributes over the graph (clarification will be updated in the paper).
>
> **Question: Comparison with GARNET [1]**
>
> Response: Thank you for the suggestion. We present the comparison on Cora, CS, and Chameleon dataset. The backbone is GCN for both, hence for Chameleon dataset we have only used 10% Metattack.
>
> | **Dataset** | **Attack**      | **GARNET** | **AdaLip** |
> |-------------|-----------------|------------|------------|
> | CS          | Metattack (10%) | 72.70      | **74.21**  |
> | CS          | Metattack (25%) | 67.74      | **72.35**  |
> | Cora        | Metattack (10%) | **80.68**  | 78.14      |
> | Cora        | Metattack (25%) | 74.81      | **76.22**  |
> | Chameleon   | Metattack (10%) | 34.81      | **36.12**  |
>
>
> **Question: Use of Projected Gradient descent**
>
> Response: By using projected gradient descent similar to [2], the parameters can be normalized after every iteration in order to have a $\kappa$-Lipschitz stable model ($\kappa$ being tuned as hyperparameter). In literature often the $\kappa$ is taken as unity. However, several studies have pointed the tradeoff in terms of expressiveness while doing such normalization [3,4]. Hence, in this work we propose an optimization based method for adaptive Lipschitz regularization. The discussion will be added in the paper.
>
> **Question: Clarification on heterophily datasets.**
>
> Response: In the experiment section, we have mentioned that for heterophily datasets we have set parameter $\beta\rightarrow 0$.
>
> **Question: Clarification of data adaptivity**
>
> Response: Since, the proposed approach have a joint method to learn GNN parameters along with graph denoising, the algorithm becomes adaptive to the data given as input.
>
> **Question: Clarification of "adaLip w/o GD" in Figure 1**
>
> Response: In Figure 1, AdaLip w/o GD means our proposed approach without using graph denoising objective ($\beta=0$).
>
> **Question: Clarification on Remark 2**
>
> Response: In Remark 2, we mean to say that low-rank approximation based graph denoising can also be used within the framework of AdaLip. Also, in figure 1 we have compared the accuracy and Lipschitz bound of different methods.
>
>
>
> [1]: Deng, C., Li, X., Feng, Z., & Zhang, Z. (2022, December). GARNET: Reduced-Rank Topology Learning for Robust and Scalable Graph Neural Networks. In Learning on Graphs Conference (pp. 3-1). PMLR.
>
> [2]: Gouk, H., Frank, E., Pfahringer, B., & Cree, M. J. (2021). Regularisation of neural networks by enforcing lipschitz continuity. Machine Learning, 110, 393-416.
>
> [3]: Zhao, X., Zhang, Z., Zhang, Z., Wu, L., Jin, J., Zhou, Y., ... & Yan, D. (2021, July). Expressive 1-lipschitz neural networks for robust multiple graph learning against adversarial attacks. In International Conference on Machine Learning (pp. 12719-12735). PMLR.
>
> [4]: Anil, C., Lucas, J., & Grosse, R. (2019, May). Sorting out Lipschitz function approximation. In International Conference on Machine Learning (pp. 291-301). PMLR.

---

> ### Author Response · Authors · 2023-11-22
> **Robustness Unit Test**
>
> **Robustness Unit Test**
>
> We perform a robustness test of our proposed method following [1]. A unit test is proposed consisting of several adaptive attacks. The attacks that are based on Vanilla GCN are considered non-adaptive. The unit test consists of poisoning attacks over a range of budget 0 to 15% of E (E=Total no. of edges). We present the results of the attacks that lead to maximum degradation in Vanilla GCN performance on Cora-ML dataset.
>
> | **Adaptive Attack** | **GCN** | **RGCN** | **ProGNN** | **AdaLip** |
> |---------------------|---------|----------|------------|------------|
> | Jaccard-GCN         | 57.60   | 56.84    | 61.67      | **71.08**  |
> | GCN-SVD             | 80.33   | 79.49    | 80.08      | **82.89**  |
> | RGCN                | 46.48   | 49.20    | 49.14      | **66.61**  |
> | ProGNN              | 38.85   | 34.61    | 33.65      | **59.61**  |
> | GNNGuard            | 58.35   | 61.16    | **67.21**  | 62.63      |
> | GRAND               | 42.81   | 41.56    | 48.65      | **63.28**  |
>
>
> Note that the hyperparameters for each model are tuned based on clean data and kept same throughout the experiment.
>
>
>
>
>
>
>
>
>
> [1]: Mujkanovic, F., Geisler, S., Günnemann, S., & Bojchevski, A. (2022). Are Defenses for Graph Neural Networks Robust?. Advances in Neural Information Processing Systems, 35, 8954-8968.

---

> > ### Comment · Reviewer_uQo7 · 2023-11-22
> > **Follow-up**
> >
> > Thanks for the detailed response and additional experiments. However, my major concerns are still valid. In particular, the new results of Robustness Unit Test are still considered as the transfer attack setting. Authors should adaptively attack their proposed model instead.
> >
> > Thus, I keep my score unchanged.

---

### Official Review · Reviewer_5YoB · 2023-10-31

**Soundness:** 1 poor
**Presentation:** 2 fair
**Contribution:** 2 fair
**Rating:** 3
**Confidence:** 3

**Summary:**

In this paper the authors propose a graph adversarial attack defense mechanism, based on the Lipschitz constant and its regularization.

The authors provide significant amount of theory and then present the experimental evaluation of their method.

**Strengths:**

The paper is easy to follow. The results look promising.

**Weaknesses:**

* The authors propose an optimisation based method. This approach requires taking the gradient of the network and then applying it to the learned weights. However, it is not promised that the network itself is a valid potential function. Therefore, I am afraid that it cannot be guaranteed that the method should converge. Therefore I believe that the theoretical guarantees are not complete as not all assumptions are provided, and also it is not clear if the experiments are conducted with a network that is a potential function. To my understanding, the authors use GCN as a backbone, which is not guaranteed to be a valid potential function. I look forward to read the authors response.

* The authors should add comparisons with recent methods such as "Robust Mid-Pass Filtering Graph Convolutional Networks"

* The authors should discuss recent findings about the evaluation of GNN robustness and conduct experiments with additional benchmarks to show the performance of the model. Please see discussion and data in "Are Defenses for Graph Neural Networks Robust?"

* The authors should provide the runtimes of the method.

**Questions:**

Please see my review

---

> ### Author Response · Authors · 2023-11-22
> **Response to Reviewer 5YoB**
>
> Thanks for your detailed review. We are going to address your concerns one by one as follows:
>
> **Question: Convergence of the algorithm**
>
> Response: In the proposed approach, we regularize the GCN loss for Lipschitz stability. There has been multitude of work proving the convergence guarantees of GCN [1,2]. Proof is presented in Appendix E.
>
> **Question: Comparison with Robust Mid-Pass Filtering Graph Convolutional Networks (Mid-GCN) [3]**
>
> Response: Thank you for the suggestion. The table presents average accuracy over ten random runs.
> | **Dataset** | **Attack**       | **Mid-GCN** | **AdaLip** |
> |-------------|------------------|-------------|------------|
> | Cora        | Metattack (25%)  | 72.89       | **76.22**  |
> | CS          | Metaattack (25%) | 69.12       | **72.35**  |
> | Cora        | Nettack (5)      | 68.56       | **74.04**  |
> | CS          | Nettack (5)      | **77.12**   | 76.19      |
>
>
> **Question: Runtime Comparison**
>
> Response: The table shows runtime comparison of different methods for 400 epochs on Cora dataset.
>
> | **Method** | **Time (s)** |
> |------------|--------------|
> | GCN        | 5.87         |
> | ProGNN     | 770.56       |
> | AdaLip.J   | 201.89       |
> | AdaLip.2   | 56.29        |
>
> [1]: Awasthi, P., Das, A., & Gollapudi, S. (2021). A convergence analysis of gradient descent on graph neural networks. Advances in Neural Information Processing Systems, 34, 20385-20397.
>
> [2]: Keriven, N., Bietti, A., & Vaiter, S. (2020). Convergence and stability of graph convolutional networks on large random graphs. Advances in Neural Information Processing Systems, 33, 21512-21523.
>
> [3]: Huang, J., Du, L., Chen, X., Fu, Q., Han, S., & Zhang, D. (2023, April). Robust Mid-Pass Filtering Graph Convolutional Networks. In Proceedings of the ACM Web Conference 2023 (pp. 328-338).

---

> > ### Author Response · Authors · 2023-11-22
> > **Robustness Unit Test**
> >
> > **Robustness Unit Test**
> >
> > We perform a robustness test of our proposed method following [1]. A unit test is proposed consisting of several adaptive attacks. The attacks that are based on Vanilla GCN are considered non-adaptive. The unit test consists of poisoning attacks over a range of budget 0 to 15% of E (E=Total no. of edges). We present the results of the attacks that lead to maximum degradation in Vanilla GCN performance on Cora-ML dataset.
> >
> > | **Adaptive Attack** | **GCN** | **RGCN** | **ProGNN** | **AdaLip** |
> > |---------------------|---------|----------|------------|------------|
> > | Jaccard-GCN         | 57.60   | 56.84    | 61.67      | **71.08**  |
> > | GCN-SVD             | 80.33   | 79.49    | 80.08      | **82.89**  |
> > | RGCN                | 46.48   | 49.20    | 49.14      | **66.61**  |
> > | ProGNN              | 38.85   | 34.61    | 33.65      | **59.61**  |
> > | GNNGuard            | 58.35   | 61.16    | **67.21**  | 62.63      |
> > | GRAND               | 42.81   | 41.56    | 48.65      | **63.28**  |
> >
> >
> > Note that the hyperparameters for each model are tuned based on clean data and kept same throughout the experiment.
> >
> >
> >
> >
> >
> >
> >
> >
> >
> > [1]: Mujkanovic, F., Geisler, S., Günnemann, S., & Bojchevski, A. (2022). Are Defenses for Graph Neural Networks Robust?. Advances in Neural Information Processing Systems, 35, 8954-8968.

---

> > > ### Comment · Reviewer_5YoB · 2023-11-23
> > > **Thanks**
> > >
> > > I thank the authors for their response, which I've read together with the other reviews.
> > > I think that they are still not sufficient to convince me on the problems found by the other reviewers and myself, and therefore I keep my original score.

---

### Official Review · Reviewer_umJp · 2023-11-01

**Soundness:** 1 poor
**Presentation:** 1 poor
**Contribution:** 1 poor
**Rating:** 3
**Confidence:** 4

**Summary:**

The paper claims to address the vulnerability of GNN to adversarial attacks. While the topic is of interest, the paper's approach and presentation leave much to be desired. The authors introduce AdaLip, an optimization-based framework, but the effectiveness of this method are questionable based on the provided content.

**Strengths:**

The paper attempts to introduce an optimization-based framework, which could be of interest if executed well.

**Weaknesses:**

1.  The paper lacks a clear and coherent structure. The introduction does not set a clear stage for the problem, and the motivation behind the proposed method is weak.

2. Notation Issues: The paper is riddled with unclear and undefined notations, which severely hampers its readability. Examples include:

$f$ in eq(2)

$\lambda$ in eq(4)

$\mathcal{X}$ in eq(6)

$\theta(x, \Delta)$ in eq(6).

$d$ in eq(7)

$\mathcal{L}, \mathcal{A}$ in Lemma 2

Furthermore, there are inconsistencies in notation usage, such as

 $\left(\theta_{(k+1)}, \phi_{(k+1)}\right)$ and $\left(\theta_{(k+1)}, \omega_{(k+1)}\right)$.

In eq(8), it writes $\min _{\theta, \phi \in \mathcal{S}_\phi}$, however this is different from eq(3).

3. Lack of Motivation for Lemmas: The relevance of certain lemmas, such as Lemma 1, is not clear. Why is it necessary, and how does it contribute to the overall narrative?


4. Unclear Statements: The paper contains several vague statements that lack clarity or justification:

"On the contrary, this research explores methods for enhancing the robustness of training across diverse architectural models by inherently minimizing the likelihood of failure, quantified through its stability coefficient."

Clarification needed: How does your approach differ in terms of "robustness of training" compared to other methods?

"The overall objective of learning a stable hypothesis $\theta$ as GNN parameters under the constraint of learning a denoised graph structure can be formulated as the general optimization problem". "The set $\mathbb{S}_\theta$ contains all possible stable hypothesis functions mapping from the domain set to the labelled set."

-Clarification needed: How exactly is a "stable hypothesis" defined in this context? It is just a combination of GNN under the constraint of a denoised graph structure.

"One effective method for enforcing Lipschitz constraints on a network during training involves normalizing the parameters of each layer using Projected Gradient Descent (PGD)."

-Clarification needed: Is there a reference here?

"Without loss of generality, the adaptive Lipschitz regularization in (9) can be equivalently replaced by a logarithmic counterpart."

-Clarification needed: Can you provide proof or justification for this equivalence?

"Lemma 2. By defining linear operators $\mathcal{L}, \mathcal{A}$ and respective adjoint operators $\mathcal{L}^{\star} \mathcal{A}^*$"

-Clarification needed: What is the objective of this lemma? How are these linear operators defined, and why do we need the transformation from (10) to (11)?

"$\Delta_{(k)}$ denote the optimal adjacency matrix corresponding to the optimal graph Laplacian $\phi_{(k)}$ at the $k^{\text {th }}$ iteration while solving (8)."

-Clarification needed: what are the iterations here?

5. Theoretical Errors:

In the derivation of Theorem 1, the initial inequality appears ambiguous. Either there are missing assumptions that need to be explicitly stated, or the derivation is flawed.

The assertion that $\left|X^{(0)}\right|_F=\sqrt{d}$ lacks justification. What is the basis for this equality?

Upon examining Lemma 1 and its accompanying proof, I am at a loss for words regarding its presentation and rigor.

I ceased my examination of the subsequent proofs due to the glaring inadequacies in the mathematical statements presented thus far.

6. Grammatical Oversights: The paper is marred by numerous grammatical errors, particularly concerning punctuation. A glaring oversight is the absence of punctuation marks following ALL equations throughout the document.

7. Disconnect Between Theory and Experiments: The paper claims that AdaLip performs well on heterophily graph datasets, yet there's no evidence or explanation supporting this claim.

8. Experimental Deficiencies: The experimental section is glaringly inadequate. Not only does it lack a comprehensive set of baselines, but the range of attacks considered is also severely limited. It is imperative to incorporate evaluations against poison and evasion attacks, as well as both white-box and black-box scenarios, and to consider both injection and modification types.

The glaring omission of a multitude of established works on Lipschitz regularization for GNNs is concerning. This oversight casts doubt on the rigor of the literature review.

Furthermore, the paper fails to report any computational costs, leaving readers in the dark about the practicality of the proposed method.

**Questions:**

Please clarify the issues raised in the weaknesses section.

In its current form, I cannot in good conscience recommend this paper for acceptance. I strongly advise the authors to rigorously revise and contemplate resubmission to a future conference.

---

> ### Author Response · Authors · 2023-11-22
> **Response to Reviewer umJp**
>
> Thanks for your detailed review. We are going to address your concerns one by one as follows:
>
> **Question: Undefined Notations.**
>
> Response: Thanks for pointing it out. We will clearly state each notation in the paper. $f_{\theta}$ in (12) represents the GNN function. $\lambda$ in (4) is a regularization parameter. $d$ in (7) represents the input feature dimension. $\chi$ is the domain set. $\theta(x,\Delta)$ is a typo; kindly consider it $f_{\theta}(x,\Delta)$, which represents GNN as a function of input feature and graph.
>
> **Question: Lemma 2 and operators $\mathcal{L}$ and $\mathcal{A}$.**
>
> Response: The detailed analysis of Lemma 2 and both operators is presented in Appendix C. Using the Lemma 2 problem (10) gets simplified as the matrix variable $\phi$, which is graph Laplacian can be replaced by the vector $\omega$.
>
>
> **Question: How does your approach differ in terms of "robustness of training" compared to other methods? How exactly is a "stable hypothesis" defined in this context?**
>
> Response: The stability of a neural network is quantified by its sensitivity toward small variations in the inputs. In literature, several approaches have been used to make the machine robust towards perturbations in the input data. The methods can broadly be summarized in three approaches:
>
> 1.) Training networks with perturbed versions of the data.
>
> 2.) Approaches where the goal is first to obtain the clean data and then train the network
>
> 3.) Studies that aim to use mathematical notions of stability.
>
> Previous methods are not mathematically proven to be robust toward input perturbations. In this paper, we have considered the notion of stability as a bounded Lipschitz constant for the graph neural network. The notion of stability using the Lipschitz property is quite popular in the analysis of non-linear systems. A system with a bounded Lipschitz constant can be easily proven robust toward input variations.
>
> Our current work focuses on providing a unified way where one can use the Lipschitz property while training the neural network without changing the overall architecture to provide robustness towards adversarial perturbations. Previous works that have enforced Lipschitz property while training have used brute-force methods where the layerwise weights are normalized [1,2]. This is the first work to formulate an optimization problem where the Lipschitz property is posed as an objective. Additionally, we have provided an iterative algorithm to solve the proposed optimization problem. Extensive experiments have demonstrated that the proposed approach is superior compared to state-of-the-art methods.
>
> **Question: Use of Projected Gradient descent**
>
> Response: By using projected gradient descent similar to [1], the parameters can be normalized after every iteration in order to have a $\kappa$-Lipschitz stable model. In literature often the $\kappa$  is taken as unity. However, several studies have pointed the tradeoff in terms of expressiveness while doing such normalization [2,3]. Hence, in this work we propose an optimization based method for adaptive Lipschitz regularization.
>
> **Question: Clarification on the proof of Theorem 1**
>
> Response: Thank you for highlighting this concern. We have modified the proof in the paper with all assumptions and details.
>
> **Question: Results on Heterophily datasets.**
>
> Response: The results are presented in Section 5.3.
>
> **Question: Use of $\log$ in (10)**
>
> Response: Thank you for bringing this to our attention. The problem of minimizing the product of functions is indeed transformed to minimize the sum of logarithms, aiming at simplifying mathematical analysis and ensuring numerical stability. However, our previous formulation did not explicitly involve taking the logarithm of the entire objective function, leading to a non-equivalent representation between equations (9) and (10). In line with the feedback from reviewer C1i6, we agree with the suggestion to directly add the log in the original definition eq (9).
>
> **Question: Meaning of iterations**
>
> Response: Since the problem (9) is a multi-block non-convex optimization problem, we devise an iterative algorithm for solving it.
>
> [1]: Gouk, H., Frank, E., Pfahringer, B., & Cree, M. J. (2021). Regularisation of neural networks by enforcing lipschitz continuity. Machine Learning, 110, 393-416.
>
> [2]: Zhao, X., Zhang, Z., Zhang, Z., Wu, L., Jin, J., Zhou, Y., ... & Yan, D. (2021, July). Expressive 1-lipschitz neural networks for robust multiple graph learning against adversarial attacks. In International Conference on Machine Learning (pp. 12719-12735). PMLR.
>
> [3]: Anil, C., Lucas, J., & Grosse, R. (2019, May). Sorting out Lipschitz function approximation. In International Conference on Machine Learning (pp. 291-301). PMLR.

---

> > ### Author Response · Authors · 2023-11-22
> > **Additional Results**
> >
> > **Question: Runtime Analysis**
> >
> > Response: The table shows runtime comparison of different methods for 400 epochs on Cora dataset.
> >
> > | **Method** | **Time (s)** |
> > |------------|--------------|
> > | GCN        | 5.87         |
> > | ProGNN     | 770.56       |
> > | AdaLip.J   | 201.89       |
> > | AdaLip.2   | 56.29        |
> >
> > **Robustness Unit Test**
> >
> > We perform a robustness test of our proposed method following [1]. A unit test is proposed consisting of several adaptive attacks. The attacks that are based on Vanilla GCN are considered non-adaptive. The unit test consists of poisoning attacks over a range of budget 0 to 15% of E (E=Total no. of edges). We present the results of the attacks that lead to maximum degradation in Vanilla GCN performance on Cora-ML dataset.
> >
> > | **Adaptive Attack** | **GCN** | **RGCN** | **ProGNN** | **AdaLip** |
> > |---------------------|---------|----------|------------|------------|
> > | Jaccard-GCN         | 57.60   | 56.84    | 61.67      | **71.08**  |
> > | GCN-SVD             | 80.33   | 79.49    | 80.08      | **82.89**  |
> > | RGCN                | 46.48   | 49.20    | 49.14      | **66.61**  |
> > | ProGNN              | 38.85   | 34.61    | 33.65      | **59.61**  |
> > | GNNGuard            | 58.35   | 61.16    | **67.21**  | 62.63      |
> > | GRAND               | 42.81   | 41.56    | 48.65      | **63.28**  |
> >
> >
> > Note that the hyperparameters for each model are tuned based on clean data and kept same throughout the experiment.
> >
> >
> >
> >
> >
> >
> >
> >
> >
> > [1]: Mujkanovic, F., Geisler, S., Günnemann, S., & Bojchevski, A. (2022). Are Defenses for Graph Neural Networks Robust?. Advances in Neural Information Processing Systems, 35, 8954-8968.

---

> > ### Comment · Reviewer_umJp · 2023-11-23
> >
> > I have read the responses and choose to maintain my current score. Thank you.

---

### Meta-Review · Area_Chair_nh4z · 2023-12-07

**Metareview:**

Summary: The submission proposes a method for making GNNs more robust by integrating graph denoising and network regularization.

+ The paper studies an important problem.
+ The optimization-based perspective is interesting.
+ The experiments, though limited, show improvements over some baselines.

- The paper lacks clarity.
- The correctness/significance of the theorectical claims is unclear.
- The experimental setup is lacking in some aspects, e.g. stronger attacks, and more recent baselines.

**Justification For Why Not Higher Score:**

- The paper lacks clarity.
- The correctness/significance of the theorectical claims is unclear.
- The experimental setup is lacking in some aspects, e.g. stronger attacks, and more recent baselines.

**Justification For Why Not Lower Score:**

N/A

---

### Decision · Program_Chairs · 2024-01-16

Reject